# Dynamic Forecasting and Operation Mechanism of Reservoir Considering Multi-Time Scales

Chengyu Han [1], Zhen Guo [2] , Xiaomei Sun [2,*] and Yuquan Zhang [3,4]

1   School of Human Settlements and Civil Engineering, Xi'an Jiaotong University, Xi'an 710049, China; xbnlkjdx06@163.com
2   School of Water Conservancy and Hydropower, Xi'an University of Technology, Xi'an 710048, China; zhen_g@stu.xaut.edu.cn
3   State Grid Xi'an Electric Power Supply Company, Xi'an 710049, China; djyjyth@163.com
4   School of Management, Xi'an Jiaotong University, Xi'an 710049, China
*   Correspondence: xiaomeisun1020@foxmail.com

**Abstract:** This paper proposes a feedback, rolling and adaptive operation decision-making mechanism for coupling and nesting of time scales. It is aimed at the change of time scale and the dynamics in the operation process, considering the relationship between operation period and multi-time scales. The key point is to integrate forecasting and operation in order to adapt to the multi-time scales dynamic change in the operation process. The operation process is divided into different time scales; forecasting and operation model method libraries are constructed, and the progressive updating and nesting mechanism are used to realize the process dynamic operation, according to the regulation period or operation period of the reservoir. Taking the Miyun Reservoir in Beijing, China as the research object, the operation mechanism is integrated into the operation process, and the complex forecasting operation and control mechanism are integrated, based on the integrated platform and using modern information technology. The forecasting and operation method uses classic different models, which can be selected based on different goals. The forecasting inflow is used as input, and the output is the water distribution plan, more importantly, the mechanism in the operation process is the key point. This is a rolling modification of the inflow process in the next stage, and the operation plan also changes accordingly. The feasibility, effectiveness, rationality and flexibility of the reservoir dynamic and adaptive operation are verified, so that the reservoir operation is dynamically changing and adapting to the changing demand. The proposed operation mechanism has scientific value and guiding significance to improve the reservoir operation theory, and it provides decision support for the actual reservoir operation and operation business.

**Keywords:** reservoir forecasting and operation; dynamic process; multi-time scales; integrated platform

## 1. Introduction

The problem of water scarcity in the 21st century is more and more serious, and there is a serious imbalance between supply and demand. It is related to the country's economy, social development and people's livelihood. As far as the water resources in China are concerned, the water resources are unevenly distributed in time and space, and contradictions in water use are prominent. In the case of serious water scarcity, the construction of water conservancy projects can redistribute water resources spatially, effectively alleviate the uneven distribution of water resources in time and space, and also alleviate the problem of regional water use [1–3]. Therefore, how to make the scientific and reasonable operation of the reservoir is the key for reservoir regulation, which directly affects the utilization of water resources. So, it is necessary to carry out research on the rational use of the reservoir operation.

The dynamic operation problem is a more common problem, which is more in line with the actual production needs. Many scholars pay attention to its research [4–6]. At present, it

has become one of the hot spots in the field of operation [7,8]. In essence, the whole process of reservoir operation is the control of reservoir discharge process in a period of time. It is a dynamic process. In this process, the inflow of the reservoir is dynamic, and the factors describing the state of the reservoir (water level, storage capacity, etc.) are also dynamic. Reservoir operation determines the control strategy or decision of the whole process of reservoir discharge in the whole dynamic process according to the objectives of the reservoir operation [9–11]. K.C. Abbaspour [12] applied an attractive procedure for obtaining model parameters in recent years to estimating hydraulic parameters in a lysimeter experiment. Stochastic dynamic programming (SDP) has been widely used in reservoir operation strategy considering runoff uncertainty [13–15]. Dariane [16] used an intelligent water drop algorithm to solve the reservoir optimal operation model. Ahmed [17] successfully used a genetic algorithm (GA) to solve the multi-objective reservoir optimal operation problem, and achieved good optimization results. Moeini [18] used fuzzy dynamic programming to solve the optimal operation problem of cascade hydropower stations. Afshar [19] used a particle swarm optimization (PSO) algorithm in the reservoir-group operation. Sharma S. [20] updated the flood forecast in real time through in-depth research, and established a set of perfect real-time operation systems in the flood season. At present, the research on static operation is mature and has practical application. However, a lot of literature [7,21–25] also pointed out that due to the uncertainty of reservoir operation, even if the static planned operation obtains optimal results, it will become infeasible due to random disturbance in actual operation, that is, "the plan can't keep up with the change". Therefore, dynamic operation becomes more and more important.

At present, scholars at home and abroad have carried out a lot of theoretical research on reservoir operation, and had many achievements. However, the decision making of the reservoir operation is an extremely complex process. The randomness of inflow runoff, the multiplicity of reservoir functions, the dynamic and real-time characteristics of the decision-making process, the limitations of mathematical models and solving technologies and the complexity of human factors make the reservoir operation problems present obvious unstructured or semi-structured characteristics. In order to combine the theoretical results with the actual application, a new reservoir forecasting and operation mechanism that can take the forecasting and operation as a whole is required in such a complicated setting. Therefore, considering the relationship between operation period and multi-time scales, this paper proposes a feedback, rolling and adaptive operation decision-making mechanism for coupling and nesting of time scales. The reservoir is dynamic during the operation process, including changes in incoming water, water volume, time scale, and water demand and so on. The operation plan is also dynamic, as long as there is a change in the real-time operation plan. Other time scale plans will also change due to the rolling adjustment of incoming water. The operation plans at different time scales are rolling nested, which is similar to a chain reaction between different layers.

## 2. Methodologies

### 2.1. Main Ideas

According to the existing forecasting mode and operation methods, aiming at the uncertainty and adaptability of dynamic changes in reservoir forecasting and operation, this paper studies the dynamic mechanism of reservoir forecasting and operation process. This mechanism reflects a double-layer adjustment. On the one hand, rolling adjustment refers to the change in water supply or operation plans at a certain moment, and subsequent predicted water supply will also change accordingly; nested adjustment refers to the nested adjustment of forecasting incoming water or operation plans between levels at different time scales. Based on the whole process of the reservoir operation cycle, the regulation is closely related to forecasting in the process. Only in each link of the operation process, mathematical models and methods are used to solve specific problems. The multi-time scales rolling nested mutual feedback mechanism is constructed in the process of forecasting and operation. On the visual integrated platform, the mechanism is realized, and a new

paradigm of reservoir forecasting and operation is formed, which provides the scientific support for reservoir forecasting and operation under changing conditions.

In the process of reservoir operation in the operation period, the dynamic changes of factors are fully considered, and the forecasting and operation are closely related in the process. According to the rolling correction of forecast results (long-term, medium-term, short-term and real-time), the corresponding operation plan is also modified, and multi-time scales are nested in the operation process. The operation plan is used to guide the whole process of the operation cycle, and the real-time operation scheme is used to guide the implementation of operation. In the working process, the operation plan and the implementation operation scheme are fed back to each other (see in Figure 1).

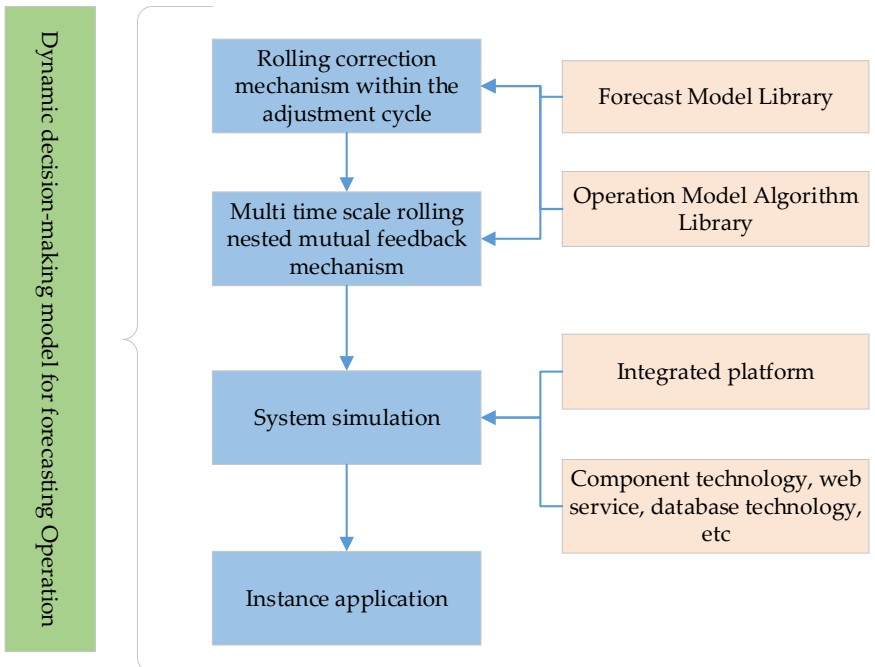

**Figure 1.** Main ideas.

During the adjustment cycle, if there is a change in incoming water at a certain moment, the forecasting of incoming water in the later period will also be reforecasted based on the latest actual incoming water. When there are new time changes, subsequent forecasting of incoming water will change and be repeated, which will lead to changes throughout the entire operation cycle. Forecasting of incoming water at other time scales will also be reforecasted based on the latest short-term results, and rolling nested reactions will occur. The operation plan will also be adjusted accordingly.

### 2.2. Rolling Correction Mechanism Mode

Reservoir forecasting and operation is an uninterrupted process, a continuous calculation process, and continuous feedback and a rolling correction process. It needs to be able to continuously provide a rolling forecast of the coming water, and the operation will also move with it.

It is assumed that the general form of the hydrological forecasting model is as follows:

$$y = f(x) \tag{1}$$

where $x$ is the sample data, $f(x)$ is the forecasting model, and it can be any model of forecasting.

(1)　The rolling process of forecast: suppose that the coming water at a certain time $t_1$ is used to forecast the coming water at the time $t_2$, and $\{x|x_0, x_1, \cdots, x_t\}$ is selected as the

sample datum of the forecast model, then when it is necessary to forecast the coming water at the time $t_3$, the sample data increase $x_{t+1}$ and becomes $\{x|x_0, x_1, \cdots, x_t, x_{t+1}\}$. At this time, in order to forecast, a new sample must be used, which can be the same length as $t_1$. At this time, the sample value $x_0$ which is far away from time $t_3$ is removed, and the new sample is $\{x|x_1, \cdots, x_t, x_{t+1}\}$, thus forming a rolling forecast.

(2) For the rolling process with measured data, it is necessary to revise the previous forecasting process. The rolling correction principle of forecast in an operation period is shown in Figure 2. The yellow part represents the historical data, the purple part represents the measured data, the orange part represents the short-term forecast results, and the red part represents the whole operation cycle. Suppose that there is a measured value $x_{t'}$ at a time $t_1$, it needs to modify the forecasting at time $t_2$, then the sample changes from the original $\{x|x_0, x_1, \cdots, x_t\}$ to $\{x|x_0, x_1, \cdots, x_{t'}\}$. At this time, the measured value is added to the sample, and the new sample is used for subsequent forecasting, so as to modify the forecasting in the process of rolling. In this way, the measured value can improve the accuracy of forecasting.

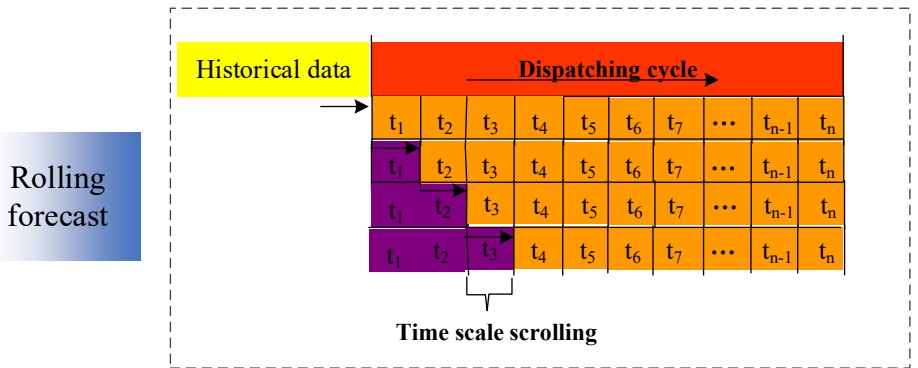

**Figure 2.** Principle of forecast rolling correction in operation cycle.

According to the rolling correction mechanism flow shown in Figure 3, suppose that the value of forecast at $t_0$ in the operation cycle is $\{Q|Q_{t1}, Q_{t2}, \cdots Q_{tn}\}$, when there is a measured value $\overline{Q}_1$ at $t_1$, the measured value $\overline{Q}_1$ needs to be added to the forecast samples, and the rolling correction forecast needs to be carried out again for the subsequent period. After rolling forecast, the forecast results of the reserved time in the cycle become $\{Q|\overline{Q}_1, Q'_{t2}, \cdots, Q'_{tn}\}$. At this time, the operation plan also changes with the change of the forecast result, and the operation plan has been performing a rolling correction, and the operation correction result is $\{Z|\overline{Z}_1, \overline{Z}_2, Z''_3, \cdots, Z''_n\}$. When there is a measured value $Q_2$ at $t_2$, the forecasting samples need to add the measured value $\overline{Q}_2$ to the forecasting samples, and the rolling correction forecast needs to be carried out again for the subsequent period. After rolling forecasting, the forecasting result of the reserved time in the cycle becomes $\{Q|\overline{Q}_1, \overline{Q}_2, Q'_{t3}, \cdots, Q'_{tn}\}$. At this time, the operation plan scheme also changes with the change of forecast, and the operation plan scheme has performed a rolling correction. The result of operation correction is $\{Z|\overline{Z}_1, \overline{Z}_2, Z''_3, \cdots, Z''_n\}$ and it has been rolling circularly. At this time, the measured value is added to the sample, and the new sample is used for subsequent forecasting, so as to roll correction forecasting in the process until the end of the operation cycle. With the continuous rolling correction of the forecast, the accuracy will also increase. The operation will change with the change of the incoming water forecast. The operation plan will be continuously modified to make the dynamic operation plan.

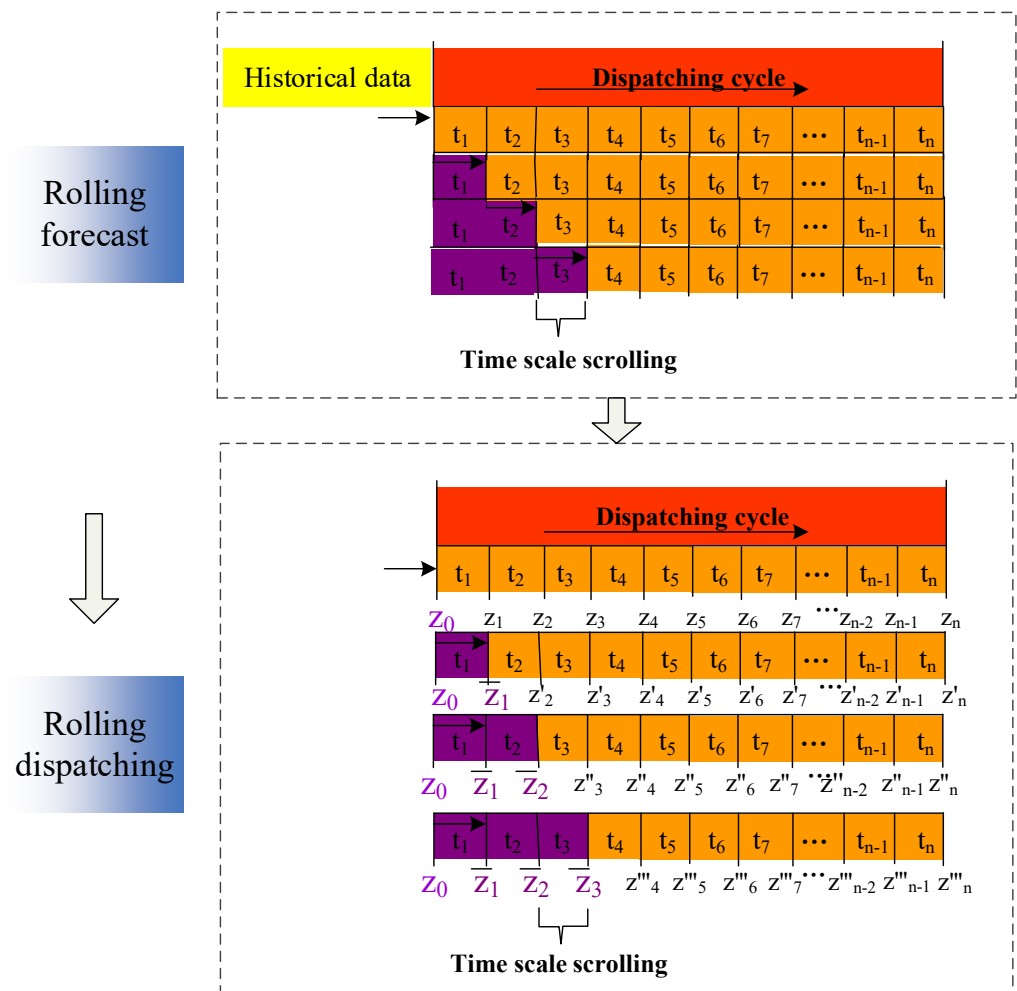

**Figure 3.** Rolling correction mechanism flow of forecast operation in operation cycle.

### 2.3. Multi-Scale Rolling Nested Mutual Feedback Mode

In this study, a mutual feedback mechanism between multi-time scales operation scheme and implementation operation scheme is established under the rolling correction mechanism to adapt to the change of conditions through process change. The working principle of the mutual feedback mechanism is shown in Figure 4. At the beginning of the operation cycle, the reservoir is operated according to the predetermined operation plan. The black arrows refer to the lengths of the time scales, and the dots refer to a long sequence of time. The yellow part represents historical data, the purple part represents measured data, the orange part represents short-term forecast results, and the red part represents the entire scheduling cycle.

(1) According to the actual inflow situation, if the real-time operation in the operation period is consistent with the operation plan, then at the end of an operation period, under the rolling nested forecasting mechanism, it is only necessary to judge whether the forecast sample data are increased. If it is increased, it will be forecasted again, and then the subsequent operation plan is formulated.

(2) The change in operation conditions leads to a deviation between the operation and the planned operation during the operation period. If an emergency occurs: an oil leakage water pollution incident occurs in the downstream of the reservoir, and the discharge flow needs to be increased for the purpose of diluting pollutants, or in management of floods, the currently implemented operation plan needs to respond to changes and make adjustments. After an operation period (as shown in $t_1$ in Figure 4), because the reservoir does not carry out the operation according to the original plan,

the follow-up operation of this period will be delayed. The operation plan should also be changed and reformulated. The actual boundary conditions (i.e., the current reservoir water level) are fed back to the upper short-term operation. Based on the rolling forecast results and the new boundary conditions, the operation model is modified, and the short-term operation plan after $t_1$ period (as shown in the yellow part of the figure, which is updated after $t_1$ period) is formulated again. The new operation plan is implemented for the next period ($t_2$ period) and the new operation plan is implemented in the whole period. According to this method, the operation period is recursively extended to medium-term operation and long-term operation until the end of the reservoir operation period.

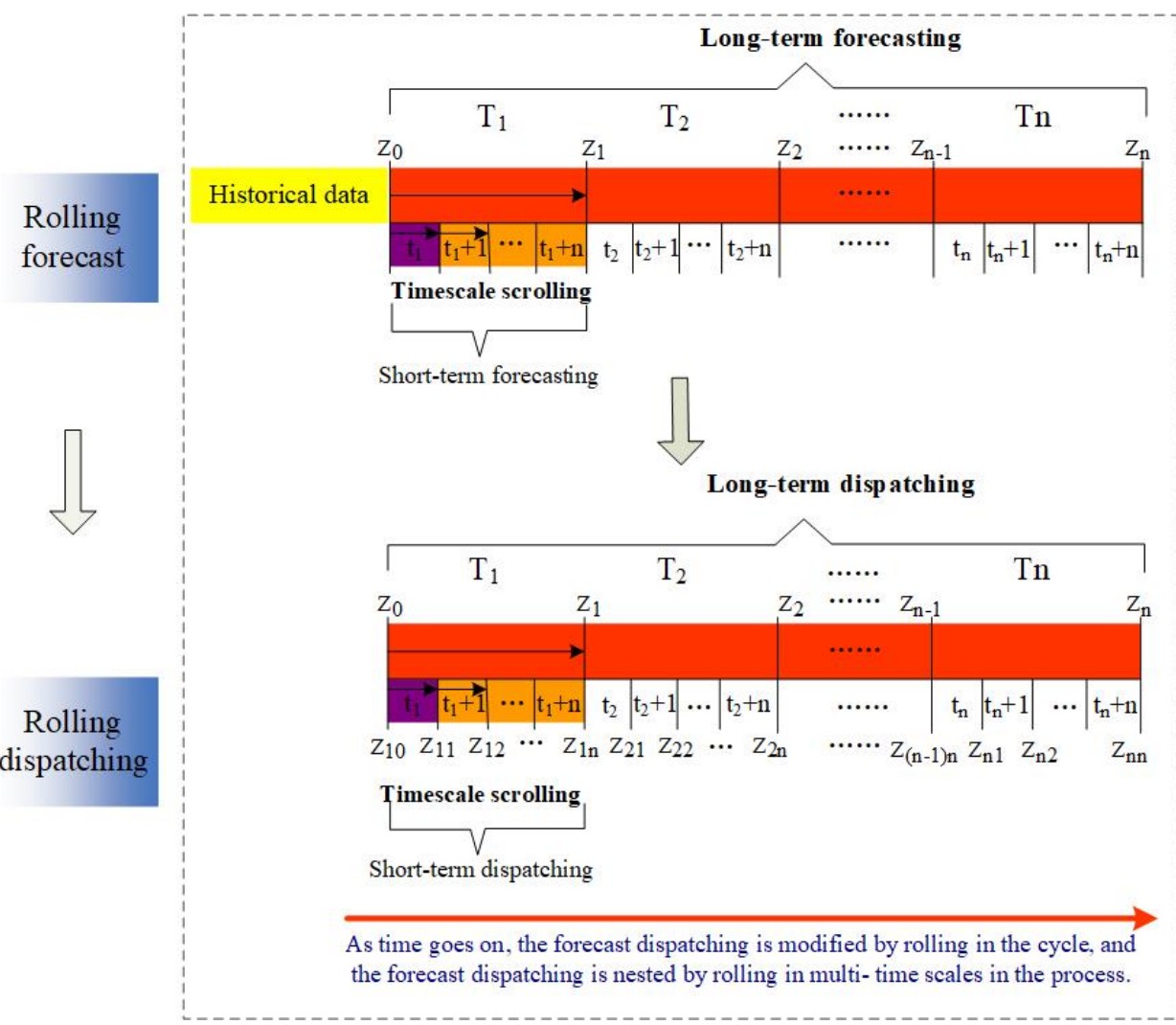

**Figure 4.** Schematic diagram of multi-time scale rolling nesting mechanism of forecast operation.

### 2.4. Forecasting and Operation Models

On the basis of existing reservoir forecasting and operation models, forecasting and operation model method libraries are constructed using information technology to serve different needs of forecasting and operation (see in Table 1), in order to make rapid calculation and analysis, such as long-term forecasting, mid-term forecasting, short-term forecasting, single objective operation, multi-objective operation, etc.

**Table 1.** Runoff forecast and operation models.

| No. | Forecasting Models | Operation Models |
|---|---|---|
| 00001 | BP | DP |
| 00002 | GM(1,1) model | POA |
| 00003 | AR_P model | Particle swarm optimization |
| 00004 | SVR model | Genetic algorithm |
| 00005 | XinAnjiang model | NSGA-II |
| 00006 | Tank madel | MOEA/D |
| 00007 | NASH unit hydrograph model | MOEA/D-AWA |
| . . . | . . . | . . . |

## 3. Implementation Means

### 3.1. Technical Support

(1)  Technical basis: The integrated platform is a platform developed based on the National 863 project in China. The platform is designed and constructed according to the requirements of the water conservancy industry standard of the People's Republic of China "technical regulations for water conservancy information processing platform" (SL538-2011). The comprehensive integration platform can integrate Web information, XML information, report information, GIS information, statistical graphics, word documents, PPT and other information, and the user layer is mainly for users with login permissions to access the platform. The overall framework is shown in Figure 5.

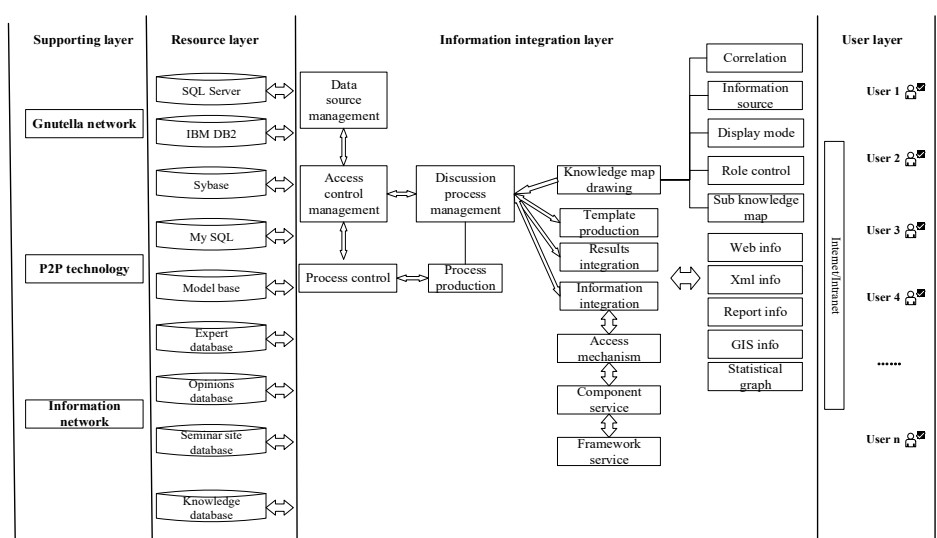

**Figure 5.** Framework design of water operation integrated platform.

(2)  Technical means: Based on the comprehensive integration platform, this paper uses the key technologies such as knowledge graph technology [26], component technology, web service technology, database technology and integrated application technology to develop and realize the dynamic decision-making system of the forecast operation process.

### 3.2. Implementation Process

In this paper, the dynamic decision-making mechanism of reservoir forecasting and operation process is established. Based on the integrated platform, the system is constructed. The hierarchical structure based on the idea of reservoir forecasting and operation is shown in Figure 6. The business and components are coupled to realize the related business requirements.

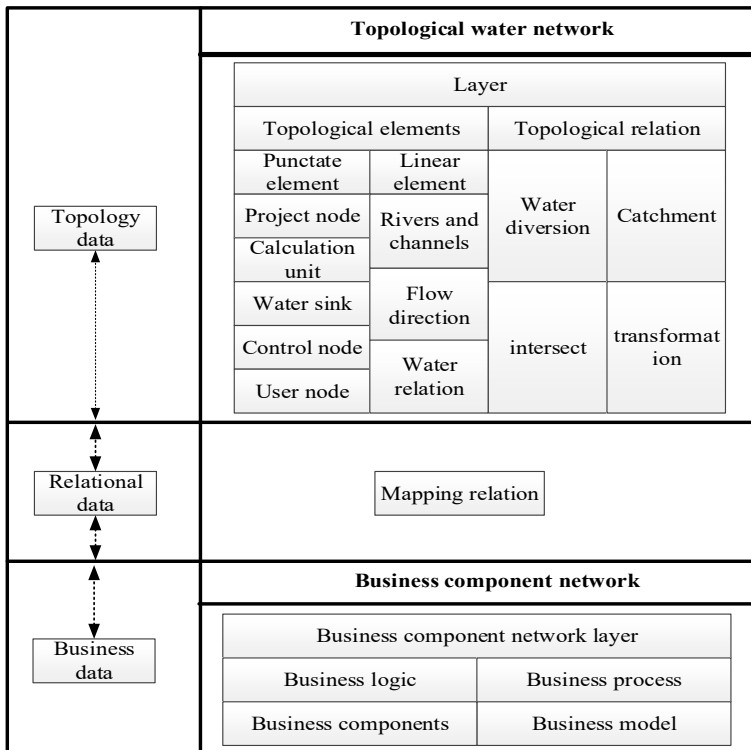

**Figure 6.** The hierarchical structure of topology and business implementation ideas.

The topological water network is composed of many nodes, and each node has a logical relationship and data flow relationship. The business of each node is supported by the components of the business component layer. Through the mapping relationship between the two, the components are customized to the nodes, and through the combined application of business components, the related business of reservoir forecasting and operation is realized.

## 4. Case Study

### 4.1. Study Area

Beijing, Tianjin and Hebei are the main platforms in China that show their strong vitality and competitiveness in the international economic system. At the same time, the region only uses 2.3% of the national land area to carry 8% of the population, and also contributes 10% of the national GDP by using less than 1% of the water. Beijing, Tianjin and Hebei are one of the regions in China and the world that are most affected by human activities and one of the regions that have the most difficulty in ensuring water safety, and with the largest water resources bearing capacity and the highest risk level [27]. Therefore, it is of great practical significance to study the water resources in this area.

The Miyun reservoir is located 13 km north of Miyun District, Beijing, in the hills of the Yanshan Mountains. It was built in September 1960. The area is 180 square kilometers, with a circumference of 200 km around Miyun Reservoir. The Miyun Reservoir is shaped like an equilateral triangle. The Miyun reservoir has a storage capacity of 4 billion cubic meters and an average water depth of 30 m, it serves as the largest and only source of drinking water supply in the capital city of Beijing. The characteristics of the Miyun reservoir are shown in Table 2. The Miyun Reservoir has two major inflow rivers, namely the Bai River and the Chao River. In the 30 years after the completion of the Miyun reservoir, it has generated huge benefits in various directions such as flood control, irrigation, urban water supply, power generation, fish farming and tourism.

**Table 2.** Miyun reservoir characters.

| Miyun Reservoir Characters | Water Level (m) | Corresponding Water Surface Area (km$^2$) | Storage Capacity ($10^8$ m$^3$) |
| --- | --- | --- | --- |
| flood level | 158.5 m | 183.6 | 41.9 |
| normal water level | 157.5 m | 179.33 | 40.08 |
| flood limit water level | 147.0 m | 137.54 | 23.38 |
| dead water level | 126.0 m | 46.154 | 4.37 |

For the accuracy of the forecasting, the historical data of incoming water were selected from 1990 to 2021 for the forecasting.

### 4.2. Models Application and System Simulation

The implementation of the rolling correction mechanism in the forecasting and operation process is shown in Figure 7. On the interface, you can see the incoming water forecasting and rolling forecasting nodes, which are supported by components in the component layer. You can see that the following blue line is also the starting condition of rolling forecasting. Based on the platform and supported by components, the rolling correction in the operation cycle can be realized on the visual flow chart. All arrows refer to the inflow and outflow of data flow. The dotted boxes refer to data nodes.

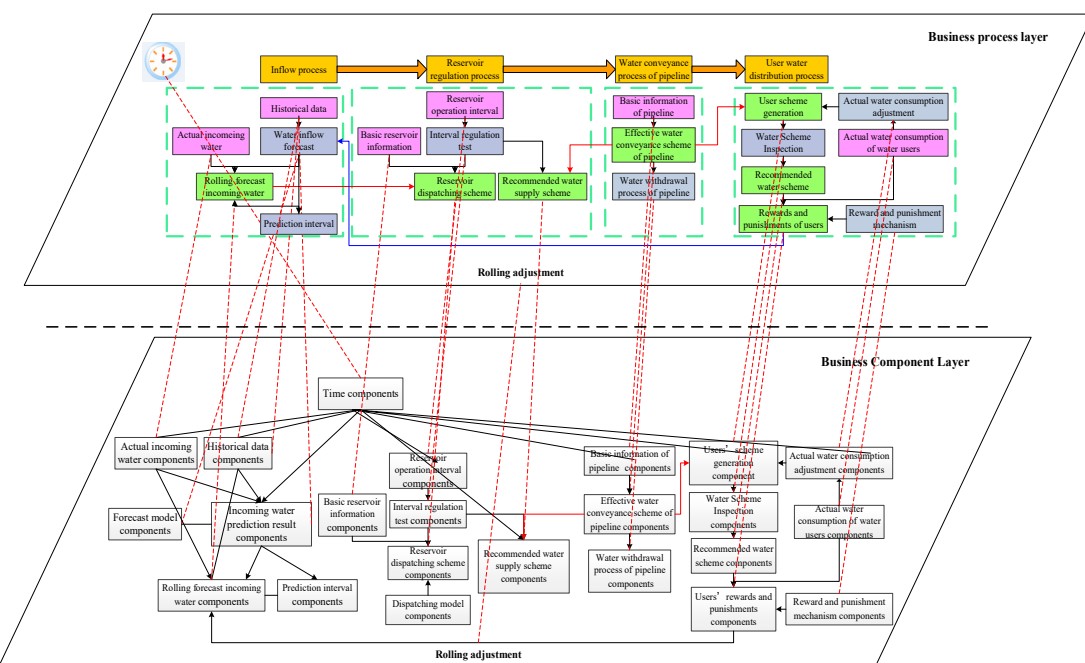

**Figure 7.** Implementation of rolling correction mechanism in forecast operation process.

### 4.3. Scenario Settings

The whole process of water adaptive regulation and control of the Miyun reservoir in China is based on the following process: water supply process → reservoir regulation process → pipeline water delivery process → user water distribution process. Through the rolling analysis of the "operation" and "distribution" process of each object, the dynamic adjustment of each link is realized, so as to achieve the process dynamic decision making of reservoir forecast and operation, and realize the global adaptive regulation.

The realization of the mutual feeding business between the mutual-time scales operation scheme and the implementation operation scheme is shown in Figure 8. On the interface, we can see that the interface is composed of two parts: the water consumption plan and the operation scheme. On the business process layer, we can see that there are two-way arrows between the planned water consumption and the operation scheme, and there are nesting

and mutual feeding relationships between them. In Figure 8, all arrows refer to the inflow and outflow of data flow, the red arrow refers to the outflow of results from the normal process of forecasting operation, and the blue arrow refers to the data flow for feedback adjustment operation.

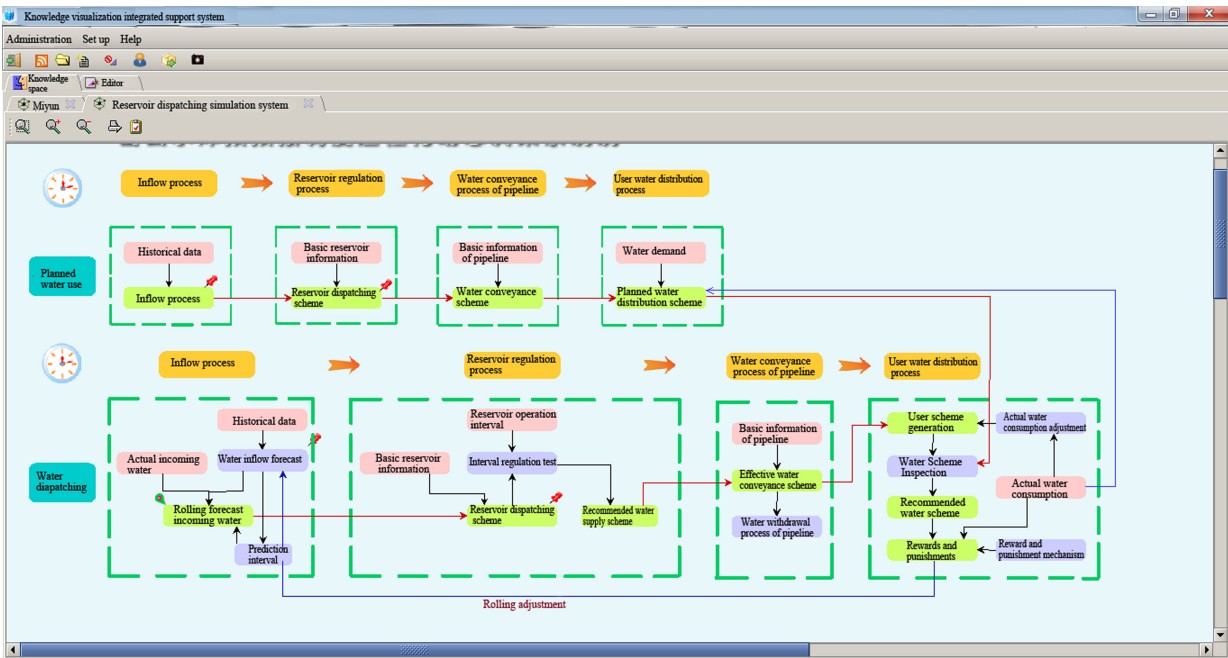

**Figure 8.** Adaptive control interface of Miyun Reservoir.

Based on the platform and supported by components, it realizes the rolling nesting mutual feed of multi-time scales in the process of the operation cycle on the visual flow chart.

### 4.3.1. Scenario 1: Rolling Forecast

The inflow process needs to be forecasted [28–30] according to the historical flow, and then the forecasted inflow process is compared with the measured inflow in the hydrological year. The rolling forecasting of the inflow process is realized by selecting the rolling forecasting time. If you choose the rolling forecast time as June 2015, the flow before June 2015 is the actual measured value, and the forecast value after that. Then the forecast is modified by rolling. The change of inflow will also lead to the change of the reservoir operation plan, so the operation is also in the state of rolling correction. Figure 9 is the rolling forecast of incoming water. In the figure, a comparison was made between the forecasted flow rate and the flow rate after rolling forecast adjustment. It can be seen that the flow rate after rolling adjustment is closer to the actual incoming water; therefore, the accuracy of the forecasting is higher.

### 4.3.2. Scenario 2: Water Distribution Plan

The operation scheme is based on the operation method library and operation target library established in advance. The operation method component and operation target component are customized on the corresponding nodes in the knowledge graph drawn on the platform. Combined with the actual operation requirements, the operation target and operation method are selected in the corresponding components to calculate the operation scheme. Reservoir regulation calculation is carried out according to its characteristic water level and storage capacity, as well as water level discharge curve and water level storage capacity curve. The pipeline calculates the water delivery capacity of the pipeline and generates the pipeline water delivery plan according to the water discharge process of the

reservoir and the attributes of the pipeline itself. The user generates the user's planned water distribution scheme according to the user's water demand and the pipeline's water delivery scheme. The operation scheme and the user's planned water distribution scheme are shown in Figure 10. In the figure, a comparison was made between the user planned water allocation and the recommended water consumption after rolling forecast adjustment. It can be seen that the recommended water consumption after rolling adjustment is closer to the actual water use, thus reflecting the applicability of this mechanism.

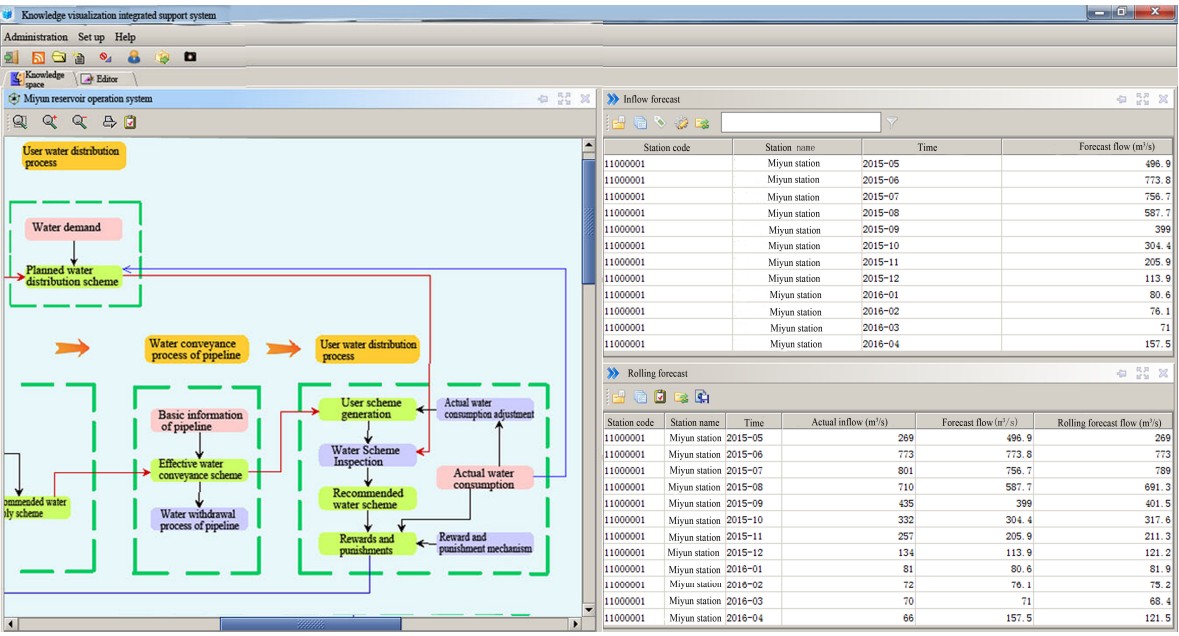

**Figure 9.** Rolling forecasting results of incoming water.

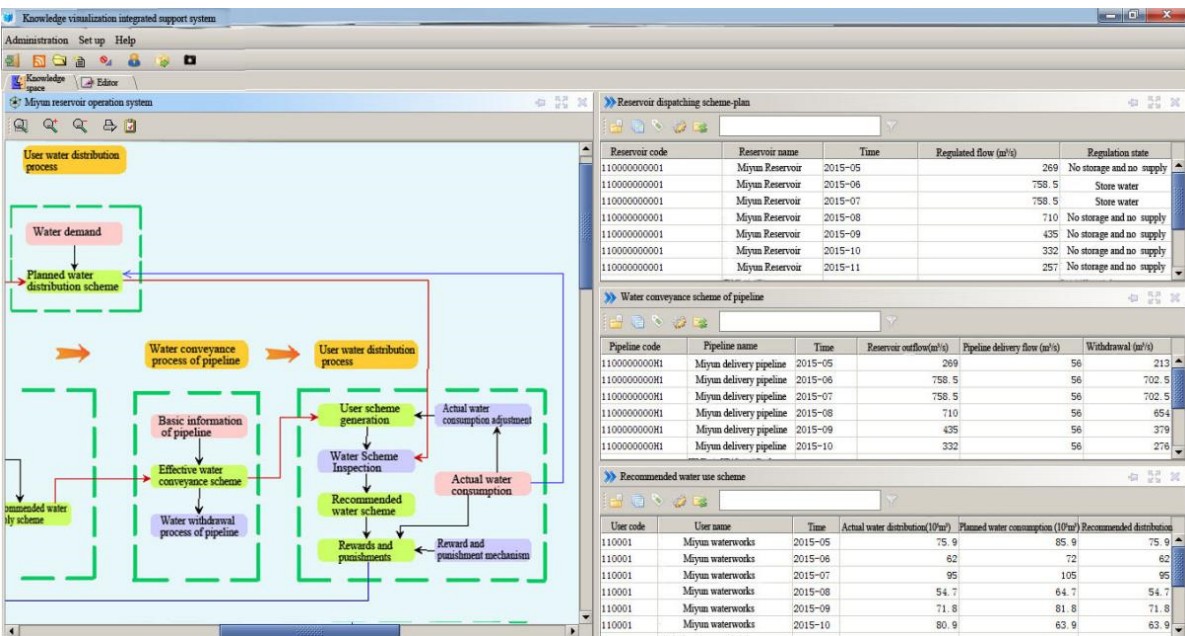

**Figure 10.** Water distribution plan of reservoir operation plan and water user plan.

### 4.3.3. Scenario 3: Water Use Plan Inspection

In the process of reservoir operation, it is necessary to conduct interval regulation inspection on the reservoir operation plan. If the discharge flow is unreasonable, the

operation regulation can be reselected or the operation range can be recalibrated. If the result is reasonable, the recommended operation scheme can be generated. Figure 11 shows the recommended water use scheme generated. The actual water use plan can be verified by the user planned water distribution plan generated above, and the excess water amount is expressed by a red bar graph. After the inspection, a recommended water use plan can be generated. Finally, according to the designated step-by-step over-plan water price as a reward and punishment mechanism, users will be punished for exceeding the planned water volume. For users with penalties, they can adjust their next water use plan, and they need to adjust the incoming water on a rolling basis to achieve dynamic rolling control of the entire water volume allocation process.

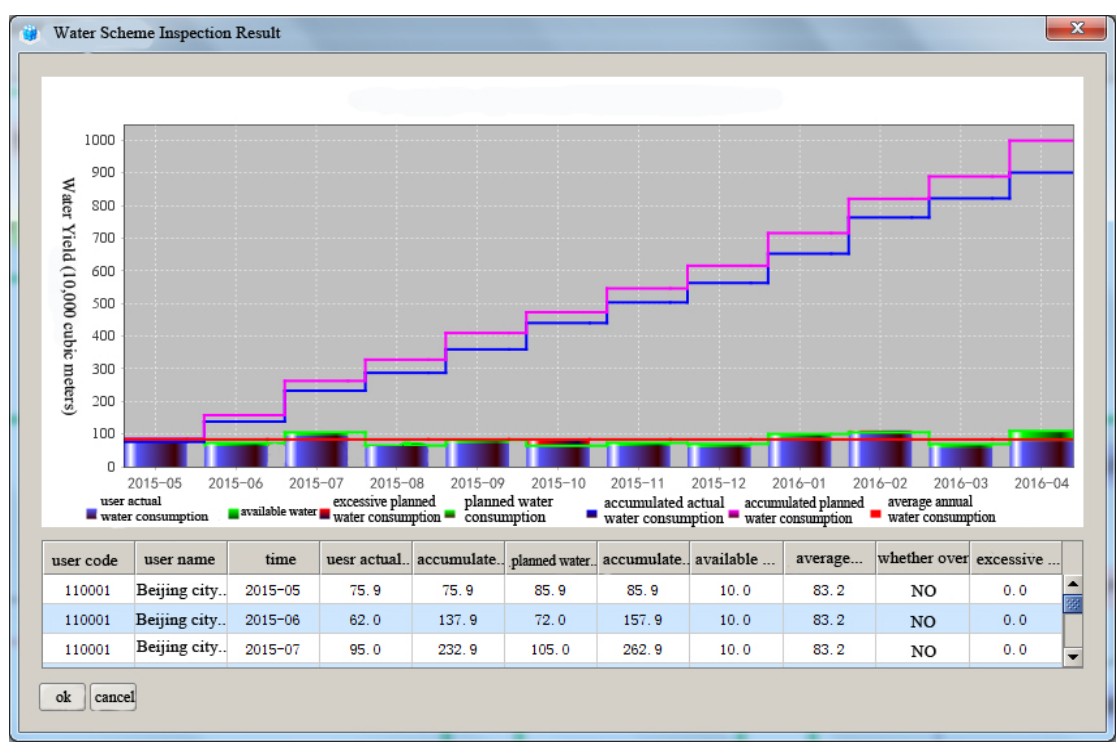

**Figure 11.** Test results of water use plan.

## 5. Results and Conclusions

The rolling correction mechanism can form a procedural operation method: in the whole reservoir operation cycle, the changing factors bring many unknowns to the operation, and also the actual operation results are not consistent with the planned scenario, and it is not clear how much the project benefits will play. In the process of operation, the time period and time scale have a great influence on the forecast operation.

The integration of reservoir forecasting, operation and decision making is realized on the platform, which can well adapt to the changes of the reservoir and achieve the purpose of dynamic operation. Through the construction of reservoir forecasting and operation process dynamic decision-making system, the rationality of the new mechanism is verified through the application of an example, which provides a new solution and means for reservoir operation business. In this paper, aiming at the time period change and the dynamic characteristics in the operation process, a process dynamic forecast operation mechanism of reservoirs is proposed, which forms a rolling correction mechanism for forecast operation in the operation period and a multi-scale rolling nested mutual feedback mechanism. Taking Miyun Reservoir in Beijing as an example, the reservoir process dynamic forecasting and operation system are carried out based on the integrated platform and computer technologies. Through the example application, the rationality of the new mechanism is verified, which provides a new solution and means for reservoir operation.

The dynamic operation of the reservoir itself is relatively complex. There are a lot of uncertainties in the operation of the reservoir. The theory and technology involved in the realization of the dynamic operation of the reservoir are relatively complex, it needs to be further solved and improved in the follow-up work. In the future, further research in this area is needed.

**Author Contributions:** Conceptualization, C.H. and Z.G.; methodology, C.H.; software, C.H., Y.Z. and Z.G.; validation, C.H. and X.S.; formal analysis, C.H. and Z.G.; investigation, C.H.; resources, X.S.; data curation, Z.G.; writing—original draft preparation, C.H.; writing—review and editing, X.S. and Y.Z.; visualization, C.H.; supervision, X.S.; project administration, X.S.; funding acquisition, X.S. All authors have read and agreed to the published version of the manuscript.

**Funding:** This research was funded by the National Natural Science Foundation of China (12272291, 11872295).

**Data Availability Statement:** The data presented in this study are available on request from the corresponding author.

**Acknowledgments:** We are grateful for the helpful and constructive review comments that helped to improve this work significantly. Therefore, we would also like to thank the reviewers.

**Conflicts of Interest:** The authors declare no conflict of interest.

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
