# Peer review of "Dynamic Forecasting and Operation Mechanism of Reservoir Considering Multi-Time Scales"

_water, doi:10.3390/w15132472_

Round 1

Reviewer 1 Report (Previous Reviewer 4)

1-      The writing of the text is not smooth and some sentences need to be rewritten. Also, some sentences do not follow each other.

2-      Line 22, the proposed model character and output should be presented in detail.

3-      Line 12, what do you mean by the "Aiming at the change of time scale and the dynamics in the operation process"

4-      Line 32, should be supported with the appropriate references. For example, see :

Anthropogenic depletion of Iran’s aquifers

Anthropogenic decline of ancient, sustainable water systems: qanats

5-      Lines 67- 75, the general character of reservoir dynamic is presented. But I don’t understand which specific character of system dynamic did you proposed? What's your research novelty in this area and what's your research difference with the available commercial model?

6-      Fig 1, the adjustment cycle and mutual feedback mechanism should be describing in more detail.

7-      The output section should be support with appropriate and more detail data.

8-      The case study data and character should be added to the manuscript.

9-      Is it not possible to do this work with the available commercial models?

10-  The frameworks of the proposed dynamic system should be introduced in order to understand the order of the system and its uncertainties should be identified.

11-  The amount of sediments entering the reservoir and its effect on reducing effective volume or issues related to the quality of the reservoir are not considered in the model.

It is strongly recommended that this manuscript be verified by a native researcher

Author Response

Responds to Reviewer 1

Comment 1: The writing of the text is not smooth and some sentences need to be rewritten. Also, some sentences do not follow each other.

 Respond: Thank you for your suggestion. Based on your suggestion, we have checked and reorganized the language of some sentences.

Comment 2: Line 22, the proposed model character and output should be presented in detail.

 Respond: Thank you for your suggestion. Based on your suggestion, I have added ' The forecasting and operation use classic different models, which can be selected based on different goals. The forecasting inflow is used as input, and the output is the water distribution plan. This is a rolling modification of the inflow process in the next stage, and the scheduling plan also changes accordingly.' to the manuscript.

Comment 3: Line 12, what do you mean by the "Aiming at the change of time scale and the dynamics in the operation process".

Respond: Thank you for your suggestion. The time scale includes long-term, medium-term, short-term, and real-time. The reservoir is dynamic during the operation process, including changes in incoming water, water volume, time scale, and water demand and so on. The operation plan is also dynamic, so the operation is a dynamic process.

 Comment 4: Line 32, should be supported with the appropriate references. For example, see :

Anthropogenic depletion of Iran’s aquifers.

Anthropogenic decline of ancient, sustainable water systems: qanats.

Respond: Thank you for your suggestion. The appropriate reference(Reference [3]Anthropogenic depletion of Iran’s aquifers.) have be added. (And, sorry, the paper ‘Anthropogenic decline of ancient, sustainable water systems: qanats’ is not found.)

Comment 5: Lines 67- 75, the general character of reservoir dynamic is presented. But I don’t understand which specific character of system dynamic did you proposed? What's your research novelty in this area and what's your research difference with the available commercial model?Respond: Thank you for your suggestion. The reservoir is dynamic during the operation process, including changes in incoming water, water volume, time scale, and water demand and so on. The operation plan is also dynamic, as long as there is a change in the real-time operation plan, other time scale plans will also change due to the rolling adjustment of incoming water. The operation plans at different time scales are rolling nested, which is similar to a chain reaction between different layers.

 Comment 6: Fig 1, the adjustment cycle and mutual feedback mechanism should be describing in more detail.

Respond: Thank you for your suggestion. We have added the more detailed description about the adjustment cycle and mutual feedback mechanism in the manuscript. ‘During the adjustment cycle, if there is a change in incoming water at a certain moment, the forecasting incoming water in the later period will also be re forecasted based on the latest actual incoming water. As new times change, subsequent forecasting of incoming water will change and be repeated, which will lead to changes throughout the entire operation cycle. Forecasting of incoming water at other time scales will also be re forecasted based on the latest short-term results, and rolling nested reactions will occur. The operation plan will also be adjusted accordingly.’

 Comment 7: Is it not possible to do this work with the available commercial models?

Respond: Thank you for your suggestion. At present, existing commercial models can perform operation, and each time scale operation is independent of each other and does not affect each other. Even if one changes, the other will not change, which is not in line with actual operation and lacks a mutual feedback mechanism. In the manuscript the system is built on the basis of the proposed mechanism, and the purpose is to achieve the proposed rolling nested adjustment mechanism that considers time scales.

 Comment 8: The frameworks of the proposed dynamic system should be introduced in order to understand the order of the system and its uncertainties should be identified.

Respond: Thank you for your suggestion. The manuscript first proposes this mechanism and its principles, and then establishes a system to achieve rolling nested adjustments at different time scales. This mechanism reflects a double-layer adjustment. On the one hand, rolling adjustment refers to the change in water supply or operation plans at a certain moment, and subsequent predicted water supply will also change accordingly; Nested adjustment refers to the nested adjustment of forecasting incoming water or operation plans between levels at different time scales. This case application verifies the rolling nested adjustment behavior of the entire mechanism in the system.

 Comment 9: The amount of sediments entering the reservoir and its effect on reducing effective volume or issues related to the quality of the reservoir are not considered in the model.

Respond: Thank you for your suggestion. The impact of these factors is not directly considered in this manuscript, but in Chapter 2.3 of the text, examples are provided to illustrate the need for the reservoir to be replanned in case of unexpected situations.

Reviewer 2 Report (Previous Reviewer 1)

No further comments

Author Response

Thank you very much for your review, and wish you all the best.

Reviewer 3 Report (Previous Reviewer 3)

This time everything is fine. The authors did a lot of effort and worked to improve the paper. 

Author Response

Thank you very much for your review, and wish you all the best.

Reviewer 4 Report (New Reviewer)

see the PDF

no

Author Response

The authors' responses can be found in the attachment: Point-by-point response to reviewers 4

Round 2

Reviewer 1 Report (Previous Reviewer 4)

The limitation mentioned in the answer to questions number 7 and 9 should be included in the manuscript

some sentences need to be improved

Author Response

Responds to Reviewer 1

Comment 7-8: The output section should be support with appropriate and more detail dataï¼›The case study data and character should be added to the manuscript.

Respond: Thank you for your suggestion. The case study data and character have been added to the manuscript. The input section has added data materials (See Chapter 4.1), in the output section, the data is stored in the system database, calculated and presented in the form of a table in the system.

Figure 9 is the rolling forecast of incoming water. In the figure, a comparison was made between the forecasted flow rate and the flow rate after rolling forecast adjustment. It can be seen that the flow rate after rolling adjustment is closer to the actual incoming water, therefore, the accuracy of the forecasting is higher.

In the Figure 10, a comparison was made between the user planned water allocation and the recommended water consumption after rolling forecast adjustment. It can be seen that the recommended water consumption after rolling adjustment is closer to the actual water use, thus reflecting the applicability of this mechanism.

Reviewer 4 Report (New Reviewer)

The authors have not responded to the main criticism of the paper.  There are two aspects that need to be addressed separately in their work: (a) accuracy of the inflow forecasts; and (b) ability of the reservoir operation model to deliver the best possible allocation based on the operational objectives.  After reading the revised version of the paper, I still do not have answers to either (a) or (b), so I am unable to assess the usefulness of the work presented in the paper.  While there are no perfect inflow forecasts, the authors could at least present the performance of the selected operational model that was applied to the Miyun reservoir, but showing its performance and ability to handle deficits in dry years.  Unfortunately, the paper continues to avoid the assessment of the performance of the reservoir operational model.  The abstract has the following sentence inserted in the revised version: “The forecasting inflow is used as input, and the output is the water distribution plan.”  I cannot see any distribution plans presented in this paper.

The paper lacks novelty, since the “rolling correction mechanism” is a standard procedure explained in other papers, the only thing that is new is the suggested name for it.  The literature mentioned in the initial review has been completely ignored.

The paper fails to evaluate how good the existing reservoir operational plan is.  This should be done by using an optimization model with the historical data and comparing the results of the operational model with the historical operation that was based on the existing plan.  If the optimization model produces significantly better results that the historical operation, that would indicate that the plan should have been changed, and it would jeopardize the conclusions made in this paper.

Figure 11 shows that the planned water use for 2015-06 was 72 units while the actual water use was 62 units.  What was the reason for this deficit, and why didn’t the “system of punishment” mentioned in the paper prevent this from happening? 

This is a poorly written paper that poses many questions for the reader, while it offers very few sensible explanations.  Inflow forecasts may be important for the high flow season, but their importance is diminished during the low flow season, when water supply comes from

The entire process presented in the paper should first be tested on the historical inflows, so as to demonstrate how well the system would work if the historical inflows were known.  Such test should state the reservoir operational objectives and provide insight into the ability of the system to meet those objectives, measured by the assessment of reliability and resilience of the system to respond to droughts when simulated over longer historic periods.  I still do not know which of the reservoir operation models listed in Table 1 was selected for the test case presented in this paper, nor am I able to assess its performance.  It is for this reason that I have requested the input and output data for the selected reservoir as accompanying data to demonstrate the validity of the work presented in this paper.  Please provide historical inflows and meteorological data series if used (precipitation and evaporation), storage capacity curves, key reservoir elevations (full supply level and dead storage level and the starting level at the beginning of the simulation), all operational objectives to be taken into account for the reservoir, simulated time step lengths and the model results (simulated storage levels, outflows and water supply to Beijing).  All of this should be provided in a spreadsheet file using a single column format for each of the variables in the list.  This request was made in the original review, but it needs to be repeated: kindly describe the system being modeled by inserting the location map, modelling schematic, clearly state the operational objectives, and provide input and output data as requested in this paragraph.

The English language used in the paper needs to be improved.  An English sentence normally starts with the subject, unlike many sentences in this paper, including the very first one “Aiming at the change of time scale and the dynamics in the operation process, considering the relationship between operation period and multi-time scales, this paper proposes a feedback, rolling and adaptive operation decision-making mechanism for coupling and nesting of time scales.”   This sentence should start with “this paper proposes . . . .”, etc.

Author Response

The author has made serious revisions and responses to the article, please see the attachment.

Round 3

Reviewer 4 Report (New Reviewer)

Kindly refer to the red italics font as the latest responses from the reviewer

Responseds to Reviewer 4

Commented 1: The authors have not responded to the main criticism of the paper.  There are two aspects that need to be addressed separately in their work: (a) accuracy of the inflow forecasts; and (b) ability of the reservoir operation model to deliver the best possible allocation based on the operational objectives.  After reading the revised version of the paper, I still do not have answers to either (a) or (b), so I am unable to assess the usefulness of the work presented in the paper.  While there are no perfect inflow forecasts, the authors could at least present the performance of the selected operational model that was applied to the Miyun reservoir, but showing its performance and ability to handle deficits in dry years.  Unfortunately, the paper continues to avoid the assessment of the performance of the reservoir operational model.  The abstract has the following sentence inserted in the revised version: “The forecasting inflow is used as input, and the output is the water distribution plan.”  I cannot see any distribution plans presented in this paper.

Respond: Thanks. In this manuscript, we mainly propose this mechanism and implement it systematically to verify its feasibility. The accuracy of prediction and the optimal allocation of reservoirs are reflected in the system, but not well reflected in the content. Perhaps the focus of the paper is on the implementation of a rolling mechanism, which can achieve rapid correction. Compared with previous forecast scheduling, it is more rapid. Previous forecast operation has independent time scales, and this mechanism in this article can achieve linkage adjustment, which is more suitable for actual operation. The water operation plan has been reflected in the system diagram.

The initial intent of the paper was to present the improved reservoir operation based on the use of better forecasting mechanism and an optimization algorithm for reservoir operation.  Based on the above response, the authors only talk about the rolling mechanism (which is nothing new) and its rapid correction.  That does not cover the two major areas of investigation that are inherent in this work: accuracy of the forecasts and the improvements in reservoir operation, which should be evaluated by comparing the model output with the historical operation.

Commented 2: The paper lacks novelty, since the “rolling correction mechanism” is a standard procedure explained in other papers, the only thing that is new is the suggested name for it.  The literature mentioned in the initial review has been completely ignored.

The paper fails to evaluate how good the existing reservoir operational plan is.  This should be done by using an optimization model with the historical data and comparing the results of the operational model with the historical operation that was based on the existing plan.  If the optimization model produces significantly better results that the historical operation, that would indicate that the plan should have been changed, and it would jeopardize the conclusions made in this paper.

Respond: Thank you for your suggestion. The literature mentioned in the initial review has been added in the manuscript, we have read the reviews, which refers to what we have learned.

Figure 9 is the rolling forecast of incoming water. In the figure, a comparison was made between the forecasted flow rate and the flow rate after rolling forecast adjustment. It can be seen that the flow rate after rolling adjustment is closer to the actual incoming water, therefore, the accuracy of the forecasting is higher.

In the Figure 10, a comparison was made between the user planned water allocation and the recommended water consumption after rolling forecast adjustment. It can be seen that the recommended water consumption after rolling adjustment is closer to the actual water use, thus reflecting the applicability of this mechanism.

Showing that water demands can be met when there is enough water would prove the applicability of any mechanism, including the standard operating policy (SOP), which boils down to “take whatever you need”.  There is nothing magic about this.  The authors should have selected a more challenging period for demonstration of their allocation mechanism, such as for example a dry period to show how the modelling system handles deficit situations over the entire irrigation season.  That would have been a more instructive example where the model results should be compared to the historic operation to demonstrate the benefits of this system.  If this comparison shows that the historic operation performed just as well as the model (or possibly even better), then the adequate conclusions should be made about the usefulness and validity of this DSS. Note that in the dry season the inflow forecasts are not relevant, the storage management is the key, while the users do not deal with storage operating rules in this paper at all.

Commented 3: Figure 11 shows that the planned water use for 2015-06 was 72 units while the actual water use was 62 units.  What was the reason for this deficit, and why didn’t the “system of punishment” mentioned in the paper prevent this from happening?

Respond: Thanks. The reason for this may be due to false reporting or different drought conditions from the previous year, resulting in this deficit. The reward and punishment system is to reward and punish first, and then reflect whether the user plan is adjusted according to the actual situation in the next year.

If the hydrologic conditions of the previous years are used as a forecast for the current year, then what is the purpose of the dynamic forecasting model presented in this paper?

Commented 4: The entire process presented in the paper should first be tested on the historical inflows, so as to demonstrate how well the system would work if the historical inflows were known.  Such test should state the reservoir operational objectives and provide insight into the ability of the system to meet those objectives, measured by the assessment of reliability and resilience of the system to respond to droughts when simulated over longer historic periods.  I still do not know which of the reservoir operation models listed in Table 1 was selected for the test case presented in this paper, nor am I able to assess its performance.  It is for this reason that I have requested the input and output data for the selected reservoir as accompanying data to demonstrate the validity of the work presented in this paper.  Please provide historical inflows and meteorological data series if used (precipitation and evaporation), storage capacity curves, key reservoir elevations (full supply level and dead storage level and the starting level at the beginning of the simulation), all operational objectives to be taken into account for the reservoir, simulated time step lengths and the model results (simulated storage levels, outflows and water supply to Beijing).  All of this should be provided in a spreadsheet file using a single column format for each of the variables in the list.  This request was made in the original review, but it needs to be repeated: kindly describe the system being modeled by inserting the location map, modelling schematic, clearly state the operational objectives, and provide input and output data as requested in this paragraph.

Respond: Thank you for your suggestion. This paper has established a model method library, which contains various model algorithms. Which model to choose is based on the actual function of the reservoir.

At least one operational model in Table 1 should be based on Linear Programming (LP), since it is the only solution algorithm that guarantees fining the global optimum, and as such it can be used to provide benchmark solutions for all other heuristic algorithms listed in Table 1.  Why was LP not included in the library?  Models such as MODSIM, WEAP, AQUATOOL, E-Water RIVERWARE, OASIS, REALM and WEB.BM are all based on linear programming, and they are widely used by the practitioners in the water resources sector.

This paper selects Miyun Reservoir as an application case. The main functions of Miyun Reservoir are flood control, irrigation, urban water supply and power generation. This time, multi-objective models and algorithms are selected for solution. The case study data and character have been added to the manuscript. The input section has added data materials (See Chapter 4.1), in the output section, the data is stored in the system database, calculated and presented in the form of a table in the system.

The whole process of water adaptive regulation and control of Miyun reservoir in China is based on the following process: water supply process → reservoir regulation process → pipeline water delivery process → user water distribution process

 I assume the “water supply process” refers to the inflow forecasting mechanism?  The term “water supply” typically refers to irrigation canal flows that may originate from dam releases which is part of the reservoir regulation process. 

 The authors have not included the data I requested that would enable independent evaluation of their reservoir operational model results:

 -        Modelling schematic of the reservoir showing its inflows, outflows and priorities among different water users

-        Elevation-area-volume curve, the hydro power plant capacity (rated head, tail water elevation, installed power, outflow vs elevation curves, pump capacity)

-        Historical inflows that can be used to test the rolling mechanism on the basis of assumed knowledge of 1, 2 or 3 days of inflow forecasts

-        Water demands for all on stream and off-stream users

-        Evaporation and precipitation in mm

-        Model solution (allocated flows and storage levels) on a daily basis for all historical years of available data from 1990 to 2021

 The above data were requested in spreadsheet format since they would enable independent model evaluation, but none were provided by the authors.  At this point, the authors pay so little attention to the model solution that they do not even bother to mention in the text which of the operational modules listed in Table 1 has been used. Also, Table 2 shows large water surface area of 179.3 km2 for the normal water level of this reservoir, which implies potential large flux caused by net evaporation (evaporation minus precipitation) that can vary throughout the year.  Which operational model was selected for reservoir operation and how is the net evaporation handled in this model?

 The authors seem to portray reservoir operation as being only dependent on the inflow forecasts.  Nothing could be further from the truth.  Inflow forecasts are very important for flood management, and not critical at all for management of droughts, with a wide range of its importance in between the floods and droughts.  The other important aspects in reservoir operation are the quality of the allocation algorithms and the ability of the model to include complex constraints on flows related to different model components. Until those issues are addressed and verifiable the evidence is provided in this paper that the selected algorithm produces high quality operational results using historical inflow series as known forecasts (for various forecasting horizons), this paper continues to have questionable value.  Kindly include the modeling schematic in the paper as previously requested, along with all requested input and output data (a web link from where the data can be downloaded should be provided in the paper) along with the results for Scenario 2 that demonstrates that you have selected an operational model that can deliver high quality solutions. The paper mentions that “The integration of reservoir forecasting, operation and decision-making is realized on the platform, . . “  but there is no sufficient evidence to judge on the quality and usefulness of this platform, since operation and decision making seem to be left up to the manual inputs by the user, rather than being driven by solutions from a powerful platform that informs the user which releases to make.

 Commented 5: The English language used in the paper needs to be improved.  An English sentence normally starts with the subject, unlike many sentences in this paper, including the very first one “Aiming at the change of time scale and the dynamics in the operation process, considering the relationship between operation period and multi-time scales, this paper proposes a feedback, rolling and adaptive operation decision-making mechanism for coupling and nesting of time scales.”   This sentence should start with “this paper proposes . . . .”, etc.

 Respond: Thanks. In this paper, some sentences in the text have been rewritten.

 The paper should be edited by a professional English language editor. The word “respond” is a verb.  The word “response” is a noun.  Also, adding references to the list at the end of the paper without quoting them in the paper is pointless.

please see the last two paragraphs in the above review.

Author Response

The reply comments are attached

This manuscript is a resubmission of an earlier submission. The following is a list of the peer review reports and author responses from that submission.

Round 1

Reviewer 1 Report

This manuscript uses dynamic forecasting and operation mechansim to study the management strategy on reservoirs, although the topic is interesting and the results are promising. I believe there are still several issues with the manuscript needs to be revised:

Firstly, the material and methods part is unclear, the author went great length to explain the dynamic mechansim, tehcnical setup, and the location and situation of the case study, which should be split up in material and methods, results and discussion. It is not easy to follow the logic, although I understand what the author want to express, it should be written in a more logical way for the readers. In the material and methods, the author should give detail on the algorithms of the forcasting methods, the location of the case study and the technical setup. In the results and discussion, the authors should report their findings in the case study and discuss more on the pros and cons of the method and mechansims, and give future outlook on the reservoir managment issue. 

Secondly, the Figures 7-10 is a little bit crude. The authors should provide data graph rather than a simple screen shot.

lastly, I believe the authors should provide some insights into the potential parameters that would influence the uncertianties of the forecasting methods, and give comparison and correlation coefficient on the predicted and actual values. 

The language should be edited throughout the manuscript, preferentially with a native speaker. Word choice and logic should be particularlly concerned. 

Author Response

Respond: Thank you for your suggestion. Based on your suggestion, in order to be suitable for a more logical way for the readers, we have reorganized the chapters of the manuscript, adjusted them to materials and methods, results and discussions, and readjusted the content of the manuscript.

In the materials and methods section, we have provided more detailed explanations on the algorithm of prediction methods(“2.4. Forecasting and operation models ‘On the basis of existing reservoir forecasting and operation models, forecasting and operation model method libraries are constructed using information technology to serve different needs of forecasting and operation, in order to make rapid calculation and analysis, such as long-term forecasting, mid-term forecasting, short-term forecasting, single objective operation, multi-objective operation, etc. Table 1 runoff forecast and operation models’”).

And we have fully enriched the location and basic information of the case(“The Miyun Reservoir is located 13 kilometers north of Miyun District, Beijing, in the hills of the Yanshan Mountains. It was built in September 1960. The area is 180 square kilometers, with a circumference of 200 kilometers around Miyun Reservoir. The Miyun Reservoir has a storage capacity of 4 billion cubic meters and an average water depth of 30 meters, making it the largest and only source of drinking water supply in the capital city of Beijing. The Miyun Reservoir has two major inflow rivers, namely the Bai River and the Chao River. In the 30 years after the completion of the Miyun Reservoir, it has generated huge benefits in various directions such as flood control, irrigation, urban water supply, power generation, fish farming, and tourism.”).

As well as we have added scenario settings, as shown in Chapter 4.3 Scenario settings). In the results and discussion, the advantages of the proposed mechanism in this manuscript were explained, and the shortcomings and future prospects were given.

Reviewer 2 Report

The article entitled " Research on Dynamic Forecasting and Operation Mechanism of Reservoir Considering Multi-time Scales " written by Han et al. proposes a process dynamic forecast operation mechanism of reservoir, focusing on the integration of forecasting and operation processes where the integration of the forecasting and operation adapting to the multi-time scales dynamic change in the operation process is aimed. A case study for Miyun Reservoir in Beijing is presented as an example, where the reservoir process dynamic forecasting and operation system are carried out based on the integrated platform and computer technologies.

In general, the study demonstrates an interesting study about dynamic forecasting and operation mechanism of reservoir process. However, I doubt the aim and scope of the paper (being on operation and forecasting mechanism as proposed rather than being on water science and technology principles) would fall under the scope of the Water Journal.

Apart from the scope, although the obtained and presented results are informative and interesting from an operations and management perspective, the methodology is also hard to follow. The paper states that the proposed operation mechanism has scientific value and guiding significance to improve the “reservoir operation theory”,  however this theory is not described at all. It is unclear if the authors are targeting to improve a theoretical research or applied research, as the example provided in their case study is confusing on this end. The presented methodology is too general, lacking specific details and lacks clarity. It is also not clear what kind of parameters are needed, why and how these would be implemented in their process. Considering the reader profile of the journal, which would be more leaning towards ecology and water resources management side, I believe the readers might find it difficult to relate the findings and the topic in general on water science and technology aspects. Based on these reasons, I believe the paper would not be suitable for publication at Water Journal.

Moderate editing of English language is required.

Author Response

Respond: Thank you for your suggestion. The purpose of manuscript research is to the change of time scale and the dynamics in the operation process, a problem-based dynamic forecasting operation mechanism of reservoir process is proposed, focusing on the integration of forecasting and operation. According to the existing forecasting mode and operation methods, aiming at the uncertainty and adaptability of dynamic changes in reservoir forecasting and operation, this paper studies the dynamic mechanism of reservoir forecasting and operation process. Based on the whole process of reservoir operation cycle, the regulation is closely related to the forecasting in the process. we have reorganized the chapters of the manuscript, adjusted them to materials and methods, results and discussions, and readjusted the content of the manuscript. In the materials and methods section, we have provided more detailed explanations on the algorithm of prediction methods. And we have fully enriched the location and basic information of the case. As well as we have added scenario settings. In the results and discussion, the advantages of the proposed mechanism in this manuscript were explained, and the shortcomings and future prospects were given.

Reviewer 3 Report

The paper deals with optimizing the operational work of the open reservoirs for water distribution. The idea is well-defined, and the methodology consists of the existing procedures and state-of-the-art. Due to the lack of energy and optimum water usage, such research is precious. The literature review gives vital insight into how to solve such issues. 

Although the research, i.e., the submitted manuscript, deserves to be published after a major revision. Here are the key points about my decision.

-authors didn't describe the water system. I.e., a size, location, position, inflow and outflow patterns, purpose, and meteorological/climate data,... Without that, I can not see the base for the methodology definition and application. 

-Authors are testing their hypothesis on only one, a short period. Such cannot be accepted for any serious research, especially publication in the Water MDPI.

-the quality of the pictures should be improved. Most of them are blurry, with a tiny font. 

It is quite a fine level of English. Only small typo errors. 

Author Response

Respond: Thank you for your suggestion. According to your suggestion, we have reorganized the chapters of the manuscript, adjusted them to materials and methods, results and discussions, and readjusted the content of the manuscript. In the materials and methods section, we have provided more detailed explanations on the algorithm of prediction methods. And we have fully enriched the location and basic information of the case, described the water system. i.e., a size, location, position, purpose. As well as we have added scenario settings. In the results and discussion, the advantages of the proposed mechanism in this manuscript were explained, and the shortcomings and future prospects were given.

Reviewer 4 Report

1-     The writing of the text is not smooth and some sentences need to be rewritten. Also, some sentences do not follow each other.

2-     I think in the title " Research on " can be removed. Also the whole title does not cover this research well.

3-     The author should have mentioned their innovation and its behind idea in the abstract and introduction section in more detail. I cannot find this research novelty in the manuscript. Also, the introduction needs some improvement to provide more information on similar works, the identified gaps, and the need for this study. The discussion section also needs some improvement regarding putting the findings of this study in the context of similar works. See the following example

"Complex dynamics of water quality mixing in a warm mono-mictic reservoir"

"Hyper-nutrient enrichment status in the Sabalan Lake, Iran"

4-     In line 79, the research question should be describing in this section. What question did you answer in this research?

5-     Line 82, what's this existing forecasting mode and operation method?

6-     Line 94, it is not clear who did you fully consider dynamic changes of factors?

7-     Figure 7 (also 8) is not clear.

8-     How can this model measure factors such as climate change or hourly fluctuations in consumption?

9-     Are the quality conditions in the reservoir that can be caused by small changes in the dam reservoir seen in this model?

10- In the results section, concrete results and the used case study should be stated in detail.

11- This model is better for applied application if it can receive its input data (time series) from online data sources.

It is suggested that the manuscript text of the article be reviewed by a native english scholar

Author Response

Respond: Thank you for your suggestion. According to your suggestion, we have made extensive revisions to this manuscript. In the title " Research on " can be removed, has been changed to “Rolling Nesting Mechanism and System Simulation for Reservoir Forecasting and Operation Considering Multi-time Scales”. We have mentioned their innovation and its behind idea in the abstract and introduction section in more detail. We have cited relevant reference in the manuscript, and see literature 24 (Roohollah Noori;Elmira Ansari;Rabin Bhattarai;Qiuhong Tang;Saber Aradpour;Mohsen Maghrebi;Ali Torabi Haghighi;Lars Bengtsson;Bjørn Kløve. Complex dynamics of water quality mixing in a warm mono-mictic reservoir . Science of The Total Environment 2021. DOI:10.1016/j.scitotenv.2021.146097.)

In the materials and methods section, we have provided more detailed explanations on the algorithm of prediction methods(“2.4. Forecasting and operation models ‘On the basis of existing reservoir forecasting and operation models, forecasting and operation model method libraries are constructed using information technology to serve different needs of forecasting and operation, in order to make rapid calculation and analysis, such as long-term forecasting, mid-term forecasting, short-term forecasting, single objective operation, multi-objective operation, etc. Table 1 runoff forecast and operation models’”).

Dynamic change factors include changes in incoming water, operation needs, operation objectives, water delivery processes, user water needs, etc.

Climate change or hourly fluctuations in consumption reflects changes in water volume, i.e. changes in incoming water.

The small changes in the dam reservoir seen in this model didn’t caused the quality conditions in the reservoir.

And we have fully enriched the location and basic information of the case. As well as we have added scenario settings, as shown in Chapter 4.3 Scenario settings). In the results and discussion, the advantages of the proposed mechanism in this manuscript were explained, and the shortcomings and future prospects were given.

Reviewer 5 Report

The authors should hire a professional language service for the write up. They should clearly express what they want to achieve. In the present state, it is pretty much incomprehensible.

The english is average but the it is difficult to understand what the authors want to say. It is pretty much meaningless.

Author Response

Respond: Thank you for your suggestion. According to your suggestion, we have made extensive revisions to this manuscript. We have reorganized the chapters of the manuscript, adjusted them to materials and methods, results and discussions, and readjusted the content of the manuscript. In the materials and methods section, we have provided more detailed explanations on the algorithm of prediction methods. And we have fully enriched the location and basic information of the case, described the water system. i.e., a size, location, position, purpose. As well as we have added scenario settings. In the results and discussion, the advantages of the proposed mechanism in this manuscript were explained, and the shortcomings and future prospects were given.